# Synergistic Effect of Carbon-Based Reinforcements on the Mechanical Properties of Cement-Based Composites

Luca Lavagna [1],* , Daniel Suarez-Riera [2] and Matteo Pavese [1]

1   Department of Applied Science and Technology, Politecnico di Torino, C.so Duca degli Abruzzi 24, 10129 Torino, Italy; matteo.pavese@polito.it
2   Department of Structural, Geotechnical and Building Engineering, Politecnico di Torino, C.so Duca degli Abruzzi 24, 10129 Torino, Italy; daniel.suarez@polito.it
*   Correspondence: luca.lavagna@polito.it; Tel.: +39-0110904598

**Abstract:** Carbon reinforcements are used to improve the mechanical properties of cement, allowing the preparation of a strengthened and toughened composite. Functionalization through a reaction with acid is necessary to guarantee both a good dispersion in water and a strong interaction with cement. Different functionalized reinforcements improve the mechanical properties of the composites in comparison with pristine cement. The use of a combination of carbon fibers, carbon nanotubes, and graphene nanoplatelets were analyzed in order to verify their synergistic effect. The use of functionalized carbon nanotubes and carbon fibers demonstrates an improvement of 71% in flexural strength and 540% in fracture energy.

**Keywords:** carbon nanotubes; carbon fibers; graphene; composites; cement

## 1. Introduction

The use of carbon-based materials in cement and concrete-based applications has only recently gained substantial research attention [1,2]. Recent studies have demonstrated that various nanofillers can be utilized to enhance the fracture toughness of cement-based materials, thereby reducing the risks associated with earthquakes, construction defects, and pressure stresses [3–8]. Furthermore, when conducting reinforcement is incorporated into a cement matrix below the percolation threshold, changes in electrical resistivity occur under applied stress, enabling real-time mechanical monitoring of structures [5,9]. Carbon nanomaterials are an obvious choice for such applications as they offer both nano-reinforcement and piezoresistive properties [10]. However, achieving optimal performance relies on two key factors: their dispersion in the cement (i.e., in the water used during cement preparation) and their interaction with the cement matrix [11,12]. The hydrophobic nature of carbon reinforcements does not guarantee their optimal interaction with the cement paste. Oxidation improves the interaction, and carbon reinforcements, especially carbon nanotubes and graphene, can serve as nucleation sites for certain hydrated phases of cement, in particular C-S-H gel ones (e.g., tobermorite) [13]. These latter phases enable the development of the matrix structure, reduce the total cement porosity and thus improve durability [14]. Furthermore, the presence of functional groups on the carbon reinforcements promotes the formation of secondary bonds, which enhance their interaction with the cementitious matrix [15]. Various methods have been proposed for dispersing nanomaterials in water, including sonication and the use of dispersants [11,16,17]. While sonication enables the fragmentation of bundles, it is insufficient to guarantee stable dispersion [18]. Dispersants effectively maintain good results but may not ensure an ideal interaction between the reinforcement and matrix, compromising the strengthening effects, conductivity, and sensing ability [19,20]. An alternative approach for achieving both a good interaction with cement and dispersion in water is the functionalization of carbon. Acidic oxidation via wet functionalization is commonly employed, as it introduces polar groups on the surface

of carbon fibers, carbon nanotubes (CNT), and graphene, enhancing the interaction with cement and dispersion in water [21–23]. However, excessive functionalization can create defects in the graphitic lattice, negatively impacting the reinforcement performance. Excessive hydrophilic groups on the CNT surface can also lead to excessive water absorption, limiting the complete hydration of cement [24]. It is worth mentioning that, in $Ca^{2+}$-rich highly alkaline water environments, a high degree of oxidative functionalization of GNPs could give rise to severe agglomeration that finally results in the worsening of mechanical performance, as observed, for example, for graphene oxide [25]. Studies suggest that a limited oxidation of carbon, ensuring the required level of oxygen-containing groups without compromising the reinforcement structure, is a favorable approach for carbon fibers and graphene [21]. In previous studies [4–6], the effect of functionalization on individual reinforcements has been analyzed to enhance their mechanical and electrical performance.

Currently in the literature, researchers are exploring the possibility of using various types of multiscale reinforcement, particularly drawing inspiration from nature [26,27]. Composites containing multiscale carbon reinforcements have been studied in polymer matrices [28], but little research has been carried out on cementitious matrices [29]. The use of different dimensions, both at the nano- and microscales or a combination of both, can improve the mechanical properties, as shown in the literature [30,31]. This study aims to investigate the synergistic effect that can be obtained by the combination of three different functionalized reinforcements in order to obtain a further improvement in the mechanical properties of cement-based composites.

## 2. Materials and Methods

Graphene nanoplatelets (GNP) grade 4 were purchased from Cheaptubes; carbon nanotubes (CNTs) NC 7000 were purchased from Nanocyl; and carbon fibers (CF) were purchased from TOHO TENAX. All the carbon-based materials were chemically oxidized following the procedure reported in previous papers [4–6].

For the oxidation of carbon nanotubes and graphene nanoplatelets, a solution of sulfuric acid was used, which is a combination of three parts sulfuric acid and one part nitric acid. For the oxidation of carbon fibers, the piranha solution was employed, which is a combination of three parts sulfuric acid and one part 30%v hydrogen peroxide.

The cement powder used in this study is ordinary Portland cement (52.5 R Ultracem) purchased from Italcementi S.p.A.

The process for creating cement composites involves dispersing carbon reinforcements in water using an ultrasonic tip for 15 min at a power level of 100 W. Subsequently, the water containing the reinforcement is mechanically stirred for several minutes, while cement powder is slowly added. The resulting mixture, prepared with a water-to-cement (w/c) ratio of 0.5 and containing 0.1% by weight of cement (bwoc) of reinforcement, is poured into dedicated molds and cured for 24 h at 85 °C in a controlled environment with 100% relative humidity. Prismatic molds measuring 20 × 20 × 80 mm are used for the cement composites, which are then subjected to mechanical testing to determine their flexural strength, fracture energy, and compression strength. The composition of the samples prepared is detailed in Table 1.

A thermo-gravimetric analysis was conducted on a TGA instrument, the Mettler Toledo 1600. Samples were heated with a constant heating ramp of 10 °C/min from 25 °C to 1000 °C. The air was supplied with a constant flow rate (50 mL/min).

The mechanical properties of the cement composites were assessed through a three-point bending test, following the ASTM C 348 standard, and controlled by crack mouth opening displacement (CMOD) as per the JCI-S-001-2003 standard. This approach allowed not only the flexural strength to be measured but also the fracture energy. The specimens were subjected to testing using a single-column Zwick-Line z050 flexural testing machine with a maximum load capacity of 50 kN and a 1 kN load cell. Following the procedure described in the literature [32], the samples were notched at their midpoint with a notch

depth of 5 mm and a width of 2 mm. CMOD was controlled at a fixed rate of 0.005 mm/min by placing an extensometer on both sides of the notch, and the span used was 65 mm.

**Table 1.** Mix design of the prepared composites.

|  | Water (g) | Cement (g) | CNTs (g) | CFs (g) | GNPs (g) |
|---|---|---|---|---|---|
| Cement | 95.2 | 190.5 |  |  |  |
| CNTs pristine | 95.2 | 190.5 | 0.19 |  |  |
| CNTs ox | 95.2 | 190.5 | 0.19 |  |  |
| CFs Pristine | 95.2 | 190.5 |  | 0.19 |  |
| CFs ox | 95.2 | 190.5 |  | 0.19 |  |
| GNPs pristine | 95.2 | 190.5 |  |  | 0.19 |
| GNPs ox | 95.2 | 190.5 |  |  | 0.19 |
| GNPs ox + CNTs ox | 95.2 | 190.5 | 0.095 |  | 0.095 |
| CFs ox + GNPs ox | 95.2 | 190.5 |  | 0.095 | 0.095 |
| CNTs ox + CFs ox | 95.2 | 190.5 | 0.095 | 0.095 |  |
| CNTs ox + CFs ox + GNPs ox | 95.2 | 190.5 | 0.063 | 0.063 | 0.063 |

After completing the flexural test, one of the two halves of each sample was utilized to determine compressive strength. Compressive tests were conducted using the same machine in load control mode, employing a 50 kN load cell, a pre-load of 20 N, and a test speed of 600 N/s. Each result for compressive and flexural strength was calculated as the average of measurements from at least four specimens.

## 3. Results and Discussion

From the thermogravimetric measurements (Figure 1), we can observe that all the oxidized carbon materials have a degradation temperature lower than their corresponding pristine ones. Specifically, regarding the degradation curve of the oxidized carbon nanotubes, the initial weight loss around 100 °C is attributed to the evaporation of condensed water [5]. The first step, ranging from 150 °C to 300 °C, is caused by the decarboxylation of the carboxylic functional groups present on the surface of the oxidized CNTs [33]. The second step, near 400 °C, corresponds to the thermal oxidation and degradation of the remaining carbon [34]. As for CFs and GNPs, only one degradation associated with the thermal oxidation of carbon was observed. The oxidized ones exhibit lower thermal resistance, primarily due to the presence of surface functional groups [35].

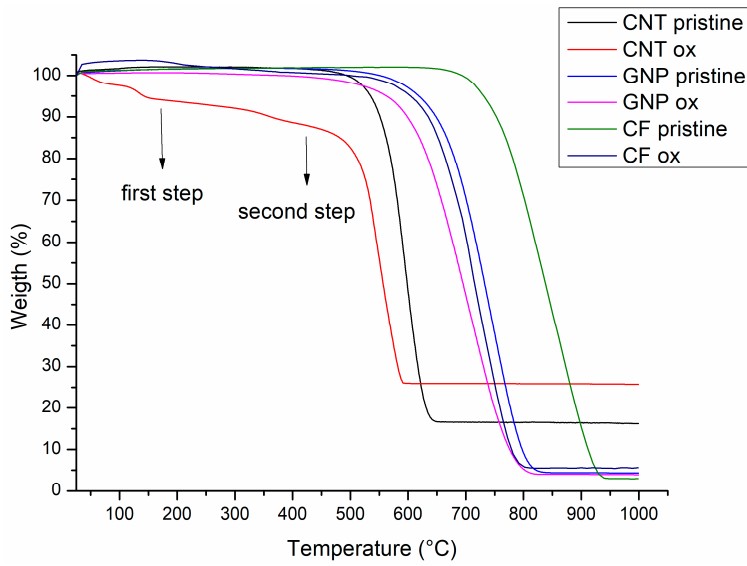

**Figure 1.** Thermogravimetric analysis of the pristine and functionalized reinforcement.

The mechanical properties of the composites containing various carbon reinforcements are shown in Figure 2 and Table 2. The non-functionalized CNTs, CFs, and GNPs exhibit no significant improvement in mechanical properties compared to their functionalized counterparts. As extensively demonstrated in the literature [11], instead, chemical functionalization not only enhances dispersion within the cement matrix but also improves the interaction with the matrix, thereby ensuring better mechanical performance. In particular, functionalized CFs exhibit a superior improvement compared to other reinforcements, primarily due to their larger size and aspect ratio [4]. Composites containing mixed reinforcements demonstrate a deterioration in compressive strength, probably due to dispersion issues in the matrix, which may also hinder the optimal hydration of the cement [36]. However, excellent results were achieved in flexural strength and fracture energy for the composites incorporating CNTs oxidized with sulfonitric acid and CFs functionalized with piranha solution, where the fracture energy increased by 540%. This improvement in flexural strength and fracture energy is related to the well-known bridging effect of CFs when they have good interaction with the cement paste [37,38]. Furthermore, the presence of carbon nanotubes functionalized with carboxyl and hydroxyl groups could promote the nucleation of cement hydrate products, thus improving the C-S-H gel structure [39,40]. The presence of GNPs seems to be a negative factor in improving mechanical properties since it reduces the beneficial effects of both CNTs and CFs, in particular regarding fracture energy. This effect is probably due to the platelet shape of GNPs that causes dispersion and re-agglomeration phenomena when mixed with fibrous-shaped reinforcements [41,42]. In Figure 2D, the flexural test curves are shown, where the toughening effect of the carbon fibers is clearly observed by the presence of a significant post-peak strength. The best sample is clearly the one containing both CFs and CNTs, with an extensive strength plateau after the peak.

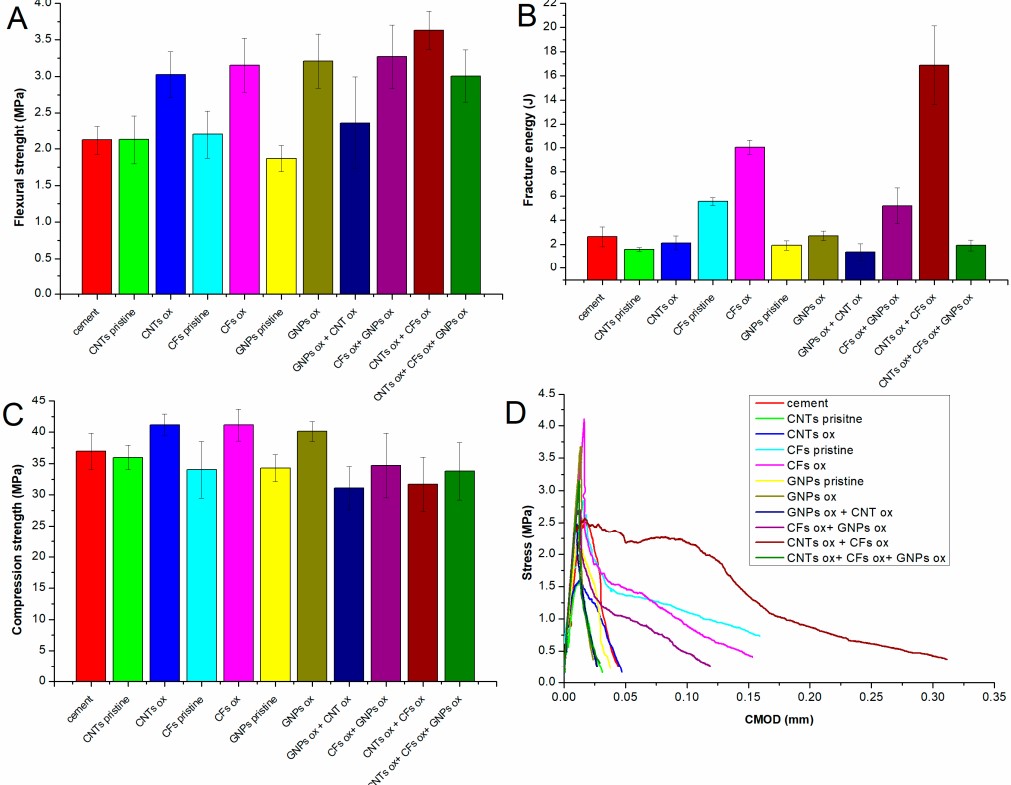

**Figure 2.** Mechanical properties of cement-based composites: (**A**) flexural strength; (**B**) fracture energy; (**C**) compression strength; (**D**) typical stress–CMOD curves for flexural strength of the 11 typology of cement composite.

**Table 2.** Sample name and the mechanical performances of the cement composites containing pristine and oxidized carbon reinforcement.

|  | Flexural Strength (MPa) * | Compression Strength (MPa) * | Fracture Energy (J) * |
|---|---|---|---|
| Cement | 2.1 ± 0.2 | 36.9 ± 2.9 | 2.6 ± 0.8 |
| CNTs pristine | 2.1 ± 0.3 | 36.0 ± 1.9 | 1.6 ± 0.2 |
| CNTs ox | 3.0± 0.3 | 41.2 ± 1.7 | 2.1 ± 0.6 |
| CFs Pristine | 2.2± 0.3 | 34.0 ± 4.5 | 5.6 ± 0.3 |
| CFs ox | 3.2 ± 0.4 | 41.1 ± 2.5 | 10.0 ± 0.6 |
| GNPs pristine | 1.9 ± 0.2 | 34.3 ± 2.2 | 1.9 ± 0.4 |
| GNPs ox | 3.2 ± 0.4 | 40.1 ± 1.6 | 2.7 ± 0.4 |
| GNPs ox + CNTs ox | 2.4 ± 0.6 | 31.1 ± 3.4 | 1.4 ± 0.7 |
| CFs ox + GNPs ox | 3.3 ± 0.4 | 34.7 ± 5.1 | 5.2 ± 1.5 |
| CNTs ox + CFs ox | 3.6 ± 0.3 | 31.7 ± 4.3 | 16.9 ± 3.2 |
| CNTs ox + CFs ox + GNPs ox | 3.0 ± 0.4 | 33.8 ± 4.6 | 1.9 ± 0.4 |

* Values are averages of at least four measurements ± SD.

To evaluate the synergistic effect of carbon reinforcements within the cementitious matrix, the following formula, inspired by the literature [43–45], was used,

$$SE = P_f - \left( \sum \frac{f_f}{f_i} P_i \right)$$

where *SE* represents the synergistic effect, $P_f$ denotes the mechanical property value, as reported in Table 2, for multiple reinforcements (either flexural strength, compression strength, or fracture energy), $P_i$ represents the value of mechanical property obtained with individual reinforcements within the matrix, $f_f$ indicates the quantity of the reinforcement used for the composites in the case of multiple reinforcements, and $f_i$ represents the quantity of the reinforcement in the case of a single reinforcement used alone within the cement matrix.

The assessment was conducted on samples in which at least two reinforcements were used. Table 3 presents the values obtained through a least-squares fitting, obtained with a Matlab^TM script, from the equations for each reinforcement, illustrating how the synergistic effect influences the individual mechanical properties.

**Table 3.** *SE* value for all the mechanical properties analyzed.

|  | Flexural Strength (MPa) | Compression Strength (MPa) | Fracture Energy (J) |
|---|---|---|---|
| CFs | 0.683 | −2.14 | 4.29 |
| CNTs | −0.160 | −5.74 | 4.42 |
| GNPs | −0.612 | −2.25 | −7.56 |

The higher and more positive the numbers, the more pronounced the synergistic effect is for that type of carbon reinforcement, while more negative numbers indicate an antagonistic effect on that specific mechanical property. Carbon fibers exhibit a synergistic effect in enhancing flexural strength and fracture energy. Carbon nanotubes contribute solely to improving fracture energy while exhibiting an antagonistic effect on compression. Graphene nanoplatelets (GNPs) appear to have a generalized antagonistic effect, especially concerning flexural strength and fracture energy.

These synergistic effect calculations confirm the observations reported above. The carbon fibers have a significant effect on enhancing the flexural strength and fracture energy, while carbon nanotubes are effective only when coupled with carbon fibers. The presence of GNPs has a generally negative effect. The compressive strength is instead negatively affected by the presence of carbon-based reinforcement.

## 4. Conclusions

This study involved the production of several cement-based composites incorporating different functionalized carbon reinforcements. As a first result, it was confirmed that the chemical functionalization or oxidation of the carbon surface enhances the dispersion and interaction of the reinforcements with the cementitious matrix. Furthermore, a synergistic effect of CNT and CF reinforcements was demonstrated, which allow almost double the flexural strength to be obtained with respect to plain cement and more than six times the fracture energy. Instead, the use of all three reinforcements together did not result in a beneficial effect on the mechanical performance. The presence of GNPs, in particular, seems to partially negate the beneficial effect of fibrous reinforcements (CNTs and CFs), probably due to dispersion and agglomeration issues. This is further confirmed by the calculation of the synergistic effect among the different reinforcement types. The synergistic effect demonstrates how the presence of carbon fibers in combination with nanotubes has a positive impact on most mechanical properties, while the use of graphene in combination with carbon fibers or nanotubes does not yield significant mechanical advantages.

**Author Contributions:** Conceptualization, L.L. and M.P.; methodology, L.L. and D.S.-R.; validation, L.L., D.S.-R. and M.P.; formal analysis, D.S.-R.; investigation, L.L.; resources, M.P.; data curation, L.L.; writing—original draft preparation, L.L.; writing—review and editing, L.L. and M.P.; visualization, L.L.; supervision, M.P.; project administration, L.L.; funding acquisition, M.P. All authors have read and agreed to the published version of the manuscript.

**Funding:** This research received no external funding.

**Data Availability Statement:** Data are available on request from the corresponding authors.

**Conflicts of Interest:** The authors declare no conflict of interest.

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
