# Peer review of "Synergistic Effect of Carbon-Based Reinforcements on the Mechanical Properties of Cement-Based Composites"

_jcs, doi:10.3390/jcs7100430_

Round 1

Reviewer 1 Report

The paper "Synergistic effect of carbon-based reinforcements on the mechanical properties of cement-based composites" by Luca Lavagna, Daniel Suarez-Riera and Matteo Pavese can be published in Journal of Composites Science after minor revisions.

Some ideas and comments are provided below in the hope that they will assist the authors in improving the quality of the article.

General comments

·         According to MDPI classification of articles (MDPI | Article Types), a communication should contain “groundbreaking preliminary results”, “cutting-edge methods or experiments” “development of new technology or materials” or “significant findings that are part of a larger study over multiple years”. Since the authors cite previously published results (Ref. 4 - 8, 10, 13), it is likely that this work falls into the last group. Seven of the overall 19 references include at least one of the authors. As a result, the authors are advised to highlight the characteristics of originality in this study. At this moment, there are no remarks to emphasize the novelty of this paper.

Suggestions

·         It is widely acknowledged that cement has a high basicity. From this point of view, acidic functionalization of carbon-based materials is logical. Some remarks related with the interaction mechanisms between i) cement and pristine carbonaceous materials and ii) cement and oxidized (functionalized) carbonaceous materials may be useful. There are interactions, chemical, physical or a combination of both. Do these reinforced cements have any chemical resistance? How sensitive are they to pH variation? How does the reinforcement impact the concrete's porosity?

·         Cement reinforcing with pristine or oxidized carbonaceous materials may show one or more of the following effects: i) cumulative (additive); ii) synergistic; or iii) antagonistic.  The scientific literature proposes several ways (formulas, equations) that allow the quantification of the synergistic impact such as synergistic effect, synergistic factor, synergistic action, synergy index etc.

(e.g.https://doi.org/10.1016/j.cemconcomp.2004.05.001, https://doi.org/10.1016/j.conbuildmat.2019.07.241, https://doi.org/10.1016/j.cemconcomp.2011.11.014)

The equation that was used to determine the synergistic effect must be provided by the authors. Furthermore, based on their field experience, in order to raise the level of originality, the authors may propose an equation that allows for a more accurate quantification of synergism.

Specific (minor) observations

·         Row 61 – a specific abbreviation is used: bwoc. Some terms are common among specialists in the subject, but they may be unclear to readers with less technical knowledge.

·         Row 113 idem – piranha solution. The chemical composition of this solution should be detailed in the Materials and Methods section (e.g. H2SO4:H2O2 ratio). 

·         Rows 118:120 – “The presence of GNPs seems a negative factor for improving mechanical properties, since it reduce the beneficial effect of both CNTs and CFs, in particular regarding fracture energy” Is this an antagonistic effect? Can it be quantified?

Reviewer 2 Report

This paper needs a major revision. 

1. In Introduction, the authors should explain why they thought the combination of three different functionalization reinforcement could obtain a further improvement of mechanical properties of cement-based composites.

2. The authors mentioned that “the initial degradation around 100 °C is attributed to the evaporation of condensed water". However, I only saw the degradation in CNT ox sample. In addition, I suggest the authors labelling the first step and the second step in Figure 1.

3. The authors mentioned that “chemical functionalization not only 105 enhances dispersion within the cement matrix but also improves the interaction with the matrix, thereby ensuring better mechanical performance.” Where is the experimental evidence like XPS, which could show the new bonding?

4. For flexural strength, fracture energy and compression strength measurements, the authors should show experimental curves and equations. In addition, where do error bars come from?

Minor revision.

Round 2

Reviewer 1 Report

  The authors successfully addressed the majority of the problems raised during the initial revision. A discussion or analysis of the reasons why GNPs have antagonistic effect on mechanical properties would be interesting, but this may be the focus of future research.

 I therefore recommend the publication of "Synergistic effect of carbon-based reinforcements on the mechanical properties of cement-based composites" by Luca Lavagna, Daniel Suarez-Riera and Matteo Pavese in Journal of Composites Science.

Reviewer 2 Report

Thanks for addressing my comments. I will suggest accepting the present version.